# Progress to Clarify How *NOTCH3* Mutations Lead to CADASIL, a Hereditary Cerebral Small Vessel Disease

**DOI:** 10.3390/biom14010127

**Published:** 2024-01-18

**Authors:** Ikuko Mizuta, Yumiko Nakao-Azuma, Hideki Yoshida, Masamitsu Yamaguchi, Toshiki Mizuno

**Affiliations:** 1Department of Neurology, Graduate School of Medical Science, Kyoto Prefectural University of Medicine, 465 Kajii-cho, Kamigyo-ku, Kyoto 602-8566, Japan; imizuta@koto.kpu-m.ac.jp (I.M.);; 2Department of Rehabilitation Medicine, Gunma University Graduate School of Medicine, Showa-machi, Maebashi, Gunma 371-8511, Japan; 3Department of Applied Biology, Kyoto Institute of Technology, Matsugasaki, Sakyo-ku, Kyoto 606-8585, Japan; 4Kansai Gakken Laboratory, Kankyo Eisei Yakuhin Co., Ltd., 3-6-2 Hikaridai, Seika-cho, Kyoto 619-0237, Japan

**Keywords:** Notch signaling, CADASIL, *NOTCH3*, EGF-like repeat, extracellular domain, granular osmiophilic material (GOM), transendocytosis, extracellular matrix protein

## Abstract

Notch signaling is conserved in *C. elegans*, *Drosophila*, and mammals. Among the four *NOTCH* genes in humans, *NOTCH1*, *NOTCH2*, and *NOTCH3* are known to cause monogenic hereditary disorders. Most *NOTCH*-related disorders are congenital and caused by a gain or loss of Notch signaling activity. In contrast, cerebral autosomal dominant arteriopathy with subcortical infarcts and leukoencephalopathy (CADASIL) caused by *NOTCH3* is adult-onset and considered to be caused by accumulation of the mutant NOTCH3 extracellular domain (N3ECD) and, possibly, by an impairment in Notch signaling. Pathophysiological processes following mutant N3ECD accumulation have been intensively investigated; however, the process leading to N3ECD accumulation and its association with canonical NOTCH3 signaling remain unknown. We reviewed the progress in clarifying the pathophysiological process involving mutant NOTCH3.

## 1. Introduction

Cerebral autosomal dominant arteriopathy with subcortical infarcts and leukoencephalopathy (CADASIL) is the most common form of hereditary cerebral small vessel disease and is characterized by recurrent stroke events, mood disturbance, and cognitive impairment [1]. The causative gene *NOTCH3* encodes the NOTCH3 receptor protein, which mediates Notch signaling. All known causative mutations of CADASIL are located in the NOTCH3 epidermal growth factor-like repeat (EGFr) domain, which is part of the NOTCH3 extracellular domain (N3ECD). Most of the mutations are cysteine-altering missense mutations, resulting in alterations in the number of cysteine residues in a certain EGFr to an odd number [2]. Notch signaling is conserved from *C. elegans* to humans, and some *Drosophila melanogaster Notch (N)* lines harbor cysteine-altering mutations in the EGFr domain, corresponding to CADASIL-causing mutations. However, animal models of CADASIL other than rodent models have not been established. This may be partly because Notch signaling is not impaired in the presence of most CADASIL-causing mutations. Many in vitro and in vivo studies have suggested that accumulation of mutant N3ECD leads to CADASIL [3]; however, the precise mechanism of N3ECD accumulation and its association with Notch signaling remain unknown. In this review, we introduce animal and cellular models of CADASIL to discuss the pathophysiology of CADASIL-causing *NOTCH3* mutations.

## 2. Overview of CADASIL

### 2.1. Cerebral Small Vessel Diseases

Cerebrovascular disease, that is, brain circulation disorder, is manifested by stroke and transient ischemic attack (TIA). Stroke is an acute neurologic injury due to brain ischemia (~80%) or brain hemorrhage (~20%) [4], which can be diagnosed by neurological examination and neuroimaging, computed tomography (CT), and/or magnetic resonance imaging (MRI). Ischemic brain injury, called infarction, is permanent. TIA is now defined as a transient episode of neurologic dysfunction caused by focal brain, spinal cord, or retinal ischemia, without acute infarction [5].

Lesions of not only large but also small vessels predispose humans to brain ischemia (Figure 1). From clinical and pathological perspectives, small vessels in the brain generally correspond to small arteries or arterioles perforating brain parenchyma. Clinical and neuroimaging characteristics of cerebral small vessel disease are small lacunar infarcts (small ischemic lesions), microbleeds, white matter lesions, and vascular dementia [6]. Most patients are sporadic, with onsets and progressions being age-dependent and lifestyle-related. Patients suffer from stroke at 65 years old (y.o.) or older and have certain conventional cardiovascular risk factors, including hypertension, dyslipidemia, diabetes, arrhythmia, and smoking [6].

In contrast, rare hereditary, monogenic small vessel diseases are typically characterized by young onset and recurrent stroke episodes without cardiovascular risk factors. To date, around eight causative genes have been identified in hereditary small vessel diseases [7], and CADASIL is the most common form worldwide, with an estimated prevalence of 2–5/100,000 [8].

### 2.2. History of Disease Concept of CADASIL

Since 1955, independent families with hereditary cerebral small vessel diseases have been reported mainly from Europe. Retrospectively, the first patient with CADASIL was identified in 1976 [1,2]. In 1993, the acronym CADASIL was proposed to designate the autosomal dominant form of cerebral small vessel disease mapped to chromosome 19 [9]. Pathological examination of autopsied brains of CADASIL patients revealed the deposition of granular material in the media of small arteries [10,11]. The deposit was generically named granular osmiophilic material (GOM) in 1995 [12] and identified as a pathological hallmark of CADASIL [13]. GOM can be detected by electron microscopic observation of areas within or adjacent to the basement membrane of vascular smooth muscle cells (VSMCs) of systemic arterioles (Figure 2); therefore, skin biopsy is useful for the definitive diagnosis of CADASIL [12].

Linkage analysis finally determined the CADASIL locus as chromosome 19p13 [14,15]. In 1996, through cDNA library screening involving the CADASIL locus, Joutel et al. characterized the *NOTCH3* gene, the human homologue of mouse *Notch3*, and identified *NOTCH3* mutations in patients [15]. As a result, CADASIL can now be definitively diagnosed through genetic testing of *NOTCH3* or detection of GOM in skin biopsy specimens, consistent with the first guideline on the diagnosis and management of cerebral small vessel diseases recently proposed by the European Academy of Neurology [16].

**Figure 2 biomolecules-14-00127-f002:**
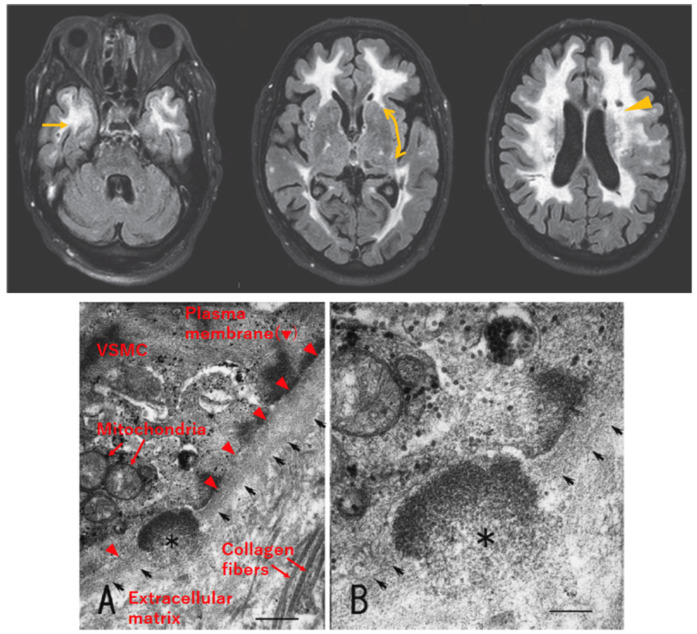
Radiological and pathological characteristics of CADASIL. Upper: Axial section of fluid-attenuated inversion recovery (FLAIR) MRI of a patient with CADASIL, reproduced with permission from Mizuno, 2012 [17], published by the Japanese Society of Neurology, with slight modification. White matter lesions are detected as hyperintense areas in deep white matter, including the temporal pole (left, arrow) and external capsule (middle, bidirectional arrow, faint signal in this patient). Lacunar infarcts are detected as punctiform low-intensity lesions (right, arrowhead). Lower: Electron micrograph of skin biopsy specimen, reproduced with permission from Mizuno et al., 2008 [18], published by the Japanese Society of Internal Medicine, with slight modification. Granular osmiophilic material (GOM, asterisk) exists within the basement membrane (arrows) of vascular smooth muscle cells (VSMCs). Labeling of cellular structures is presented in red in (**A**). Scale bar indicates 410 nm in (**A**) and 200 nm in (**B**).

### 2.3. Clinical Characteristics of CADASIL

The typical natural history of CADASIL was concisely described in previous reviews [1,2,19]. Briefly, as the disease progresses, migraine with aura, stroke/TIA, mood disturbance, apathy, motor disability, and dementia occur sequentially in this order. Migraine often occurs as the initial symptom, at 6–48 (mean: 30) y.o. [2]. Although most patients have few or no vascular risk factors, including hypertension, hyperlipidemia, diabetes, and smoking, they suffer from stroke/TIA repeatedly. The mean age at onset of the initial ischemic event is around 49 y.o. [2], which is younger compared with that in sporadic stroke patients [20]. Neurological symptoms include motor palsy, pseudo-bulbar palsy, sensory disturbance, parkinsonism, and seizure. Additional symptoms are psychiatric disturbances, including severe mood disturbance (20% of patients) and apathy (40% of symptomatic patients) [2]. Executive dysfunction is detected by neuropsychological tests in 89% of patients, even in the early stages of the disease [21]. Cognitive impairment progresses, and dementia is present in 90% of patients prior to death [1].

White matter lesions (WMLs) are present as symmetrical hyperintense signals on T2-weighted or fluid-attenuated inversion recovery (FLAIR) MRI (Figure 2) in the early stages of the disease, preceding ischemic events [2]. WMLs often involve the periventricular region, external capsule, and/or temporal pole, and the lesion volume increases with aging. Especially, WMLs in the temporal pole show high sensitivity (90%) and high specificity (90%) for the diagnosis of CADASIL [19]. Regarding WMLs in the external capsule, their sensitivity for the diagnosis of CADASIL is also high, at 90%; however, their specificity is low, at 50% [19]. Other findings related to small vessel disease, lacunar infarcts and microbleeds, can also be detected in CADASIL patients [2].

It was reported that WMLs were detected in a 21 y.o. individual, who was an asymptomatic member of a CADASIL family [22]. In addition, it was reported that GOM was detected in a 19 y.o. patient, although it was not stated whether he/she was symptomatic [23]. These findings suggest that the imaging and pathological characteristics of CADASIL can be detected as early as around 20 y.o., being earlier than clinical symptoms.

### 2.4. Pathology of CADASIL

Histopathological findings in CADASIL patients include the thickening of vascular walls, luminal stenosis, and degeneration of VSMCs and pericytes, resembling those of sporadic small vessel diseases [13,24,25]. As described above, a CADASIL-specific pathological finding is GOM deposition within or adjacent to the basement membrane of VSMCs or pericytes detected on electron microscopic observation (Figure 2). GOM deposits often localize in infoldings of VSMCs and in close contact with the plasma membrane but separated from the plasma membrane by a thin electron-lucent halo; however, they are sometimes observed distant from the plasma membrane [26,27,28]. NOTCH3 consists of an extracellular domain (N3ECD) and intracellular domain (N3ICD) (see Section 3.1). Using antibodies against N3ECD or N3ICD, light and electron microscopic observations showed that N3ECD, not N3ICD, is accumulated in vessel walls and colocalized with GOM deposits [26,28,29].

### 2.5. Genetics of CADASIL

#### 2.5.1. Autosomal Dominant Inheritance

CADASIL is inherited in an autosomal dominant manner, and most patients have heterozygous mutations. Patients with homozygous mutations have also been reported. Although some of them showed a more severe phenotype than heterozygous patients in the same family, the range of phenotypic severity in homozygous patients is considered to be similar to that in heterozygous patients [30].

#### 2.5.2. Typical Cysteine-Altering Mutations in EGFr of NOTCH3

The *NOTCH3* gene spans 42 kb and consists of 33 exons. Nearly 300 *NOTCH3* mutations related to CADASIL have been reported [21,31,32,33]. All known mutations are located in the NOTCH3 EGFr domain (see Section 3.1), encoded by exons 2–24. Each EGFr normally contains six cysteine residues that are considered to form three pairs of disulfide bonds to stabilize the protein conformation and possibly form disulfide bonds with cysteines from the neighboring repeat domain. Most pathogenic mutations are cysteine-altering missense ones, resulting in a change in the number of cysteine residues in a certain EGFr to an odd number, disrupting the structure of not only the repeat the variant is within but also the neighboring repeat domain. Consequently, protein misfolding and/or interaction with another molecule via an unpaired cysteine can occur [34,35], which may lead to the formation of GOM deposits. The extracellular accumulation of misfolded N3ECD may be due to exocytosis, aggregation on the plasma membrane, and/or impairment of transendocytosis (see Section 5.2.2), although the precise mechanism remains unknown.

#### 2.5.3. Atypical Mutations in NOTCH3

A few cysteine-sparing mutations in *NOTCH3* were reported in CADASIL patients [36]. Among them, the substitution of arginine at amino acid position 75 of NOTCH3 to proline (p.Arg75Pro) is relatively common in Japanese and Korean patients with CADASIL [37,38,39], with convincing evidence of pathogenicity based on a segregation study and detection of GOM by skin biopsy [18]. Patients with p.Arg75Pro showed a lower frequency of temporal pole WML and later onset compared with patients with cysteine-altering mutations, suggesting an association of p.Arg75Pro with atypical and mild phenotypes [38,39,40].

p.Arg544Cys is a cysteine-altering mutation, pathologically confirmed by detecting GOM [41], but it is atypical because it localizes not in EGFr but between EGFr 13 and 14. Patients with p.Arg544Cys are frequently identified on Jeju Island in Korea and in Taiwan. Totals of 90.3 and 70.5% of CADASIL patients on Jeju Island and in Taiwan, respectively, were reported to have p.Arg544Cys [42,43]. In addition, patients with homozygous p.Arg544Cys have been repeatedly reported [41]. p.Arg544Cys is correlated with a low frequency of WMLs in the temporal pole and late age at onset [43], similar to p.Arg75Pro.

#### 2.5.4. EGFr Location–Phenotype Correlations

As for typical cysteine-altering mutations, some genotype–phenotype correlation studies have been reported [39,44]. However, because of numerous mutations and various genotype distributions among populations, replication studies are considered difficult. Recently, the effect of the location of cysteine-altering mutations on the CADASIL phenotype has been suggested [45]. In CADASIL patients, typical cysteine-altering mutations are mostly located in EGFr 1–6 among the 34 EGFr [8]. In contrast, cysteine-altering *NOTCH3* variants in a general population, revealed by recent large-scale genetic variation databases, accumulated in EGFr 7–34 [8]. Total frequencies of variants of the general population were 0.4–11.7/1000, being approximately 100-fold higher than the prevalence of CADASIL, at 2–5/100,000, suggesting that *NOTCH3* variant carriers in general populations may be asymptomatic or preclinical CADASIL patients, or the variant may be of low penetrance [8]. Based on the difference in the variation hotspot between CADASIL patients and general individuals, Rutten et al. compared the disease severity between patients with EGFr 1–6 mutations and those with EGF 7–34 ones. They found that cysteine-altering mutations in EGFr 1–6 were associated with a more severe phenotype, earlier age at onset of stroke, and shorter life expectancy, than those in EGFr 7–34 [45]. An earlier age at onset of stroke in EGFr 1–6 was also noted in a Japanese study, suggesting that the effect of the mutation location on the phenotype is a consensus finding in CADASIL worldwide.

Recently, Hack et al. suggested a three-tiered EGFr domain stratification, high risk (EGFr 1–6, 8, 11, and 26), medium risk (EGFr 9–10, 12–15, 17, 25, 27, and 32), and low risk (EGFr 16, 18–20, 23–24, 28–31, and 33), according to intensive analysis of the literature on CADASIL and population genome databases [46]. A comparison of MRI findings, stroke risks, and pathological findings among these groups supported this classification [46].

## 3. NOTCH3 and Notch Signaling

### 3.1. Protein Structure of NOTCH3 Receptor

Notch family members and their ligands are single-pass transmembrane proteins. Notch is encoded by a single gene, *Notch* (*N*), in *Drosophila*, whereas there are four *Notch* family genes, *NOTCH1-4*, in mammals. The predominant form of cell surface Notch protein is a heterodimer of NECD and Notch intracellular domain (NICD) in both *Drosophila* and mammals [47]. NOTCH3, as well as other NOTCH family proteins, is S1-cleaved in the trans-Golgi network and then produces a non-covalent heterodimer of N3ECD and N3ICD [48]. The molecular sizes of full-length NOTCH3, N3ECD, and N3ICD are 290, 230, and 95 kDa, respectively [49]. N3ECD includes 34 EGFr and Lin12-Notch repeats (LNRs). N3ICD contains the transmembrane domain (TMD), RBPjκ association module (RAM), nuclear localization sequences (NLSs), ankyrin repeats (ANK), and proline/glutamate/serine/threonine motifs (PEST) (Figure 3) [48,50]. Mammalian Notch ligands are Jagged-1,2 (Jag1, Jag2) and Delta-like-1, 3, 4 (Dll1, Dll3, Dll4), which correspond to fly Serrate and Delta, respectively. Ligand–receptor interaction is mediated by the DSL (Delta, Serrate, LAG-2) domain of the ligand and EGFr domain of the receptor. *Drosophila* Notch contains 36 EGFr. An aggregation assay using *Drosophila* S2 cells transfected with deletion constructs of *Notch* and those transfected with *Serrate/Delta* showed that EGFr 11–12 of fly Notch were sufficient and necessary for ligand binding [51]. Through sequence comparison, EGFr 10–11 of human NOTCH3, corresponding to fly Notch EGFr 11–12, were predicted to mediate ligand binding [52], which was supported by a Notch signaling activity assay using human 293T cells transfected with the *NOTCH3* construct lacking EGFr 10–11 [53].

### 3.2. Notch Signaling Process

The Notch signaling process is complex (Figure 4). In the heterodimeric Notch receptor, the membrane-tethered NICD is kept inactivated by LNR of the Notch extracellular domain (NECD). Following ligand–receptor binding, via DSL of the ligand and EGFr of NECD, NECD is detached from NICD to be transendocytosed to a ligand-presenting cell [58]. The remaining NICD is activated by a sequential S2- and S3- proteolytic process. The membrane-bound “a disintegrin and metalloproteinases” (ADAM) proteases, mainly ADAM10 and ADAM17, participate in juxtamembrane S2-cleavage of NICD. Then, intramembrane S3-cleavage by *γ*-secretase occurs to allow NICD to leave the membrane. In the nucleus, the transcription regulator RBPj*κ* associates with the co-repressor and represses promotor activity. After S3-cleaved NICD enters the nucleus via NLS, NICD interacts with RBPj*κ* via the RBPj*κ* association module (RAM) and ankyrin repeat (ANK), and then the co-repressor is swapped with the co-activator, resulting in activation of transcription of the target genes, including HES1 and HEYL [48,59].

### 3.3. Localization of NOTCH Family Members and Their Ligands in Vessels

An immunohistochemical study of an autopsied specimen showed that NOTCH3 was expressed in VSMCs but not detected in endothelial cells [26]. The expression profiles of Notch family members and their ligands in blood vessel cells have been intensively analyzed in the mouse embryo, pre- and post-natal mouse brain [60], and retina [61]. Hofmann and Iruela-Arispe summarized the expression profiles as follows: Notch1, Notch3, Jag1, Jag2, and Dll1 are expressed in VSMCs, whereas Notch1, Notch4, Dll1, Dll4, and Jag1 are expressed in endothelial cells [62]. The NOTCH3-JAG1 interaction is focused on in CADASIL research, because Jagged1 is expressed in both endothelial cells and VSMCs, and the co-expression of JAG1 and NOTCH3 was detected in adult humans [53]. Notch signaling in the crosstalk between VSMCs and endothelial cells is considered to play an important role in VSMC maturation [63]. Endothelial or VSMC-specific deletion in mice strongly suggests that NOTCH3-JAG1 interaction occurs between endothelial cells and VSMCs and also between neighboring VSMCs. Mice with endothelial-specific deletion of Jag1 are embryonic-lethal, showing markedly decreased expression levels of VSMC markers but intact expression levels of endothelial markers [64]. Tamoxifen-induced conditional knockout of endothelial Jag1 in adult mice caused a loss of Jag1 expression and also decreased expressions of Notch3 and its downstream molecules in neighboring VSMCs [65]. Similarly, mice with VSMC-specific deletion of Jag1 are perinatal-lethal, showing a deficit in VSMC development [66], and the conditional knockout of VSMC-specific JAG1 in adults led to reduced expression of myosin light-chain kinase and an impairment in the arterial contractile function [67].

### 3.4. Evaluation of Notch Signaling

For cellular experiments to quantify Notch signaling, the human embryonic kidney cell line HEK293 and mouse fibroblast cell line NIH 3T3 are often used because of their low-level expression of endogenous NOTCH receptors. A ligand-binding assay can be performed by adding a soluble ligand-Fc fusion protein to a medium of cultured cells expressing the NOTCH receptor to be quantified by flow cytometry or visualized by immunostaining [58,68]. To evaluate ligand-induced Notch signaling activity, cells expressing the Notch receptor (and luciferase reporter, if necessary) should be exposed to an immobilized ligand [69] or co-cultured with ligand-expressing cells [68]. Notch signaling activity can be evaluated by quantification of transcripts of the target genes or by the Notch reporter system using the HES1 promoter-luciferase construct or tandem repeats of the RBPjκ (also called CSL: CBF-1, Suppressor of Hairless, LAG-1) binding site-luciferase construct [70].

## 4. Strategy to Clarify Pathological Mechanism of CADASIL

This review focuses on basic research to clarify the pathophysiology of CADASIL. The two major viewpoints of research are as follows: the NOTCH3 signaling process and protein accumulation or aggregation. Researchers have used various materials, including human materials, animal models, cell cultures, and peptides, in order to clarify the pathophysiology of CADASIL (Table 1).

### 4.1. Mouse Models of CADASIL

#### 4.1.1. Transgenic Mice

A CADASIL-like phenotype is not replicated in either the overexpression or knockout of wild-type *NOTCH3* in mice [25]. To date, R90C [71], R169C [72], R182C [73], C428S [74], C455R [75], and R1031C [75] *Notch3* transgenic mice, which harbor the CADASIL-causing mutations p.Arg90Cys, p.Arg169Cys, p.Arg182Cys, p.Cys428Ser, p.Cys455Arg, and p.Arg1031Cys, respectively, have been reported [25,76,77] (Table 2). Among the six mutations, pArg90Cys, p.Arg169Cys, and p.Arg182Cys are relatively frequent, whereas p.Cys428Ser, p.455Arg, and p.Arg1031Cys are rare in CADASIL patients [44]. The R169C transgene is a rat P1-derived artificial chromosome (PAC) containing the *Notch3* gene locus [72,78], and the R182C transgene is a human bacterial artificial chromosome (BAC) containing *NOTCH3* and flanking genes [73,79]. Therefore, expression profiles of the transgenes of R169C and R182C show endogenous patterns. Transgene constructs of other mice are *Notch3* cDNA driven by the VSMC-specific *SM22α* promoter (Table 2).

These transgenic mice successfully showed N3ECD accumulation and/or GOM deposition. However, a pathological white matter lesion was seen in only one model, R169C (line 88) [72], and cerebrovascular dysfunctions were reported in two models, R90C and R169C [72,78,80] (Table 2). Lacunar infarction and neurological symptoms, such as motor deficit, have never been reproduced in transgenic mouse models [25].

R169C mice showed the highest expression level of the transgene relative to endogenous *Notch3*, at 400%, and exhibited the earliest progression of pathological change, N3ECD accumulation at 1–2 months, GOM deposition at 5 months, and impaired cerebrovascular autoregulation at 5 months [72] (Table 2). R182C (line 350) also showed a high expression level of the transgene, at 350%, phenotypes from the early stage, N3ECD accumulation at 6 weeks, and GOM deposition at 5–6 months but failed to show cerebrovascular dysfunction [79].

**Table 2 biomolecules-14-00127-t002:** *NOTCH3* knock-in or transgenic mice.

Mouse (Line)	Mutation (EGFr)	Transgene Expression ^1^	Brain Pathology	Vascular Physiology
Knock-in				
*Notch3R170C/R170C* [78,81]	p.Arg169Cys (EGFr4)	not applicable	N3ECD accumulation at 4 months [78]; GOM deposition at 20 months [81]	Decreased passive diameter of isolated posterior cerebral arteries at 4 months [78]
Human *NOTCH3* cDNA driven by murine *SM22a* promoter				
*mN3+/+*; *TghN3 (WT)* (line 46) [82]	Wild-type	73%	No N3ECD accumulation; no GOM deposition	ND ^2^
*mN3+/+*; *TghN3(R90C)* (line Ma) [71,80,82]	p.Arg90Cys (EGFr 2)	86%	N3ECD accumulation and GOM deposition at 12 months [82]	Decreased flow-induced dilatation and increased pressure-induced myogenic tone in tail caudal arteries at 10–11 months [80]
*mN3+/+*; *TghN3 (C428S)* (line 10) [74]	p.Cys428Ser (EGFr 10)	150%	N3ECD accumulation and GOM deposition at 8 months [74]	ND
PAC ^3^ containing genomic locus of rat *Notch3* [72,78]				
*TgNotch3R169C* (line 88)	p.Arg169Cys (EGFr 4)	400%	N3ECD accumulation at 1–2 months; GOM deposition at 5 months; white matter lesion (numerous vacuoles and loss of compact myelin with disorganized fibers) at 18–20 months [72]	Reduction of resting CBF ^4^ in gray matter (at 11–12 months) and white matter (at 18–20 months) [72]. Impaired cerebrovascular autoregulation at 5 months and attenuated functional hyperemia at 5–6 months [72]. Decreased increment of distensibility and decreased passive diameter in isolated posterior cerebral arteries at 6 and 2 months, respectively [78]
*TgNotch3R169C* (line 92)		200%	N3ECD accumulation and GOM deposition [25]	ND
*TgNotch3WT* (line 129)	Wild-type	400%	No N3ECD accumulation: no GOM deposition up to 20 months [72]	No impairment of resting CBF, cerebrovascular autoregulation, or functional hyperemia [72].
BAC ^5^ containing genomic locus of human *NOTCH3* [73,79]				
*tgN3MUT* (line 350)	p.Arg182Cys (EGFr 4)	350%	N3ECD accumulation detected at 6 weeks: GOM deposition detected at 5–6 months [73]; no white matter lesion detected [73]	No functional deficit in CBF [79].
*tgN3MUT* (line 200)		200%	N3ECD accumulation detected at 3 months [73]	ND
*tgN3MUT* (line 150)		150%	N3ECD accumulation detected at 5 months [73]	ND
*tgN3MUT* (line 100)		100%	N3ECD accumulation detected at 12 months [73]	No functional deficit in CBF [79].
*tgN3WT*	Wild-type	100%	No N3ECD accumulation; no GOM deposition up to 20 months [73]	No functional deficit in CBF [79].
Human *NOTCH3* cDNA, *SM22*-Cre mediated conditional knock-in into *ROSA26* locus [75]				
*NOTCH3C455R*	p.Cys455Arg (EGFr 11)	ND	GOM deposition at 6 months [75]	ND
*NOTCH3R1031C*	p.Arg1031Cys (EGFr 26)	ND	GOM deposition at 12 months [75]	ND

^1^ Expression level of transgene relative to endogenous *Notch3*. ^2^ ND: not described ^3^ P1-derived artificial chromosome. ^4^ CBF: cerebral blood flow. ^5^ Bacterial artificial chromosome.

#### 4.1.2. Knock-In Mice

Of the two *Notch3* knock-in mice reported, one carrying p.Arg142Cys, corresponding to p.Arg141Cys in human NOTCH3, failed to develop any phenotype related to CADASIL [83]. In contrast, the other carrying p.Arg170Cys, corresponding to p.Arg169Cys in human *NOTCH3*, showed both the pathological and clinical characteristics of CADASIL [81]. p.Arg170Cys knock-in mice displayed GOM deposits and cerebral small vessel pathology, including thrombosis, microbleeds, and microinfarction, and also neurological symptoms including ataxia and paresis [81]. The phenotype was not observed in all animals analyzed. Of the 73 knock-in mice, brain pathology and motor defects were observed in 17 (23%) and 9 (12%), respectively, suggesting incomplete penetrance [81].

### 4.2. Cellular Model of CADASIL

#### 4.2.1. Human VSMC Cells

To investigate cellular function predisposing to CADASIL, rather than replicate the disease phenotype, some studies used partially immortalized VSMCs established from the umbilical cords of newborns whose mothers had CADASIL [69,84,85]. Jin et al. identified the gene encoding platelet-derived growth factor receptor β (PDGFRβ), a key molecule for vascular development and homeostasis, as a novel Notch signaling target gene [69]. Panahi et al. reported an increased expression of the *tumor growth factor β (TGFβ)* gene in association with a decreased proliferation rate of CADASIL VSMCs [85]. On the other hand, an increased growth rate was observed in a primary VSMC culture prepared from skin biopsy samples by Neves et al. [86]. The reason for this discrepancy remains unresolved.

#### 4.2.2. iPSC-Derived Mural Cells

Recently, Yamamoto et al. developed mural cells (VSMCs and pericytes) derived from induced pluripotent stem cells (iPS cells) of patients as a cell model of CADASIL [87]. Using the cells, they successfully reproduced previously reported CADASIL-related pathological findings. They identified increased intracellular immunoreactivity of N3ECD and N3ECD-positive deposits on the plasma membrane, in agreement with their previous findings in autopsied brains [28]. In addition, some deposits were positive for latent-transforming growth factor beta-binding protein-1 (LTBP-1) or high-temperature requirement A1 (HTRA1), which were reported as components of GOM [88,89] (see Section 6.2). The cells also showed an irregular distribution of actin filaments and increased immunoreactivity of PDGFRβ, in agreement with previous reports [90,91].

### 4.3. In Vitro Aggregation Assay

To investigate the pro-aggregation property of mutant NOTCH3, a single particle assay involving scanning for intensely fluorescent targets (SIFT) was developed [34,92]. When overexpressed, full-length N3ECD tends to be incorrectly unfolded in cells and is barely secreted in media. To obtain the unfolded recombinant N3ECD peptide of a sufficient dose from culture media, the peptide should be as short as EGFr 1–5 [34]. De novo multimer formation in a mixture of green or red fluorescent peptides can be detected by SIFT visualization. Aggregation was observed in mutant EGFr 1–5 and a mix of mutant and wild-type EGFr 1–5 but not in wild-type EGFr 1–5 [34]. The pro-aggregation abilities of some cysteine-sparing mutations, p.Arg75Pro and p.Asp80Gly, were shown by this analysis, supporting their pathogenicity [92]. Recently, Lee et al. reported structural changes in the EGFr 1–3 peptide using a gel mobility shift assay. They found an upshifted band in all the cysteine-altering mutants studied. They focused on substitution at Arg75 and found that p.Arg75Pro, p.Arg75Cys, and p.Arg75Gly peptides were upshifted, whereas substitutions from Arg75 to 16 other amino acids were not [93].

### 4.4. Drosophila Melanogaster and CADASIL

#### 4.4.1. Drosophila Melanogaster and Human Neurological Diseases

Two major categories of neurological disorders are neurodegenerative and cerebrovascular ones. *Drosophila melanogaster* models of human neurodegenerative diseases, including Alzheimer’s disease, Parkinson’s disease, repeat expansion diseases, and amyotrophic lateral sclerosis (ALS)/frontotemporal lobar degeneration (FTLD), have been established [94,95,96]. Using fly models, the high-throughput screening of modifier genes and candidate drugs is possible [97]. On the other hand, it is difficult to create *Drosophila* models of human cerebrovascular diseases, mainly because of differences in the circulatory system between insects and mammals [98]. Flies have an open circulatory system; that is, the body cavity is filled with hemolymph circulated by a contractile dorsal tube called the heart. Although a *Drosophila* model of human cardiac disease is available [99], that of vascular disease is challenging.

#### 4.4.2. Drosophila Notch Alleles Mimicking CADASIL-Causing Mutations

As described above, cysteine-altering mutations in a certain EGFr of *NOTCH3* lead to a single phenotype, CADASIL, in humans. Some *Drosophila Notch* mutations have the same characteristics, but their phenotypes are not uniform (Table 3 and Table 4). Notch signaling activity includes lateral inhibition, inductive signaling, and asymmetric cell division [100]. In *Drosophila* wings, *Notch* loss-of-function alleles cause thicker wing veins and scalloping (notching) in the wing margin. In contrast, *Notch* gain-of-function alleles, *Abruptex*, cause abrupt (shortened) wing veins and large wings [101]. All *Abruptex* mutations localize in EGFr 24–29, called the Abruptex domain. Of the *Abruptex* alleles, homozygous lethal alleles, known as *lethal-Abruptex*, were cysteine-altering mutations. Fryxell et al. hypothesized that human CADASIL corresponds to fly Abruptex because, at that time, known cysteine-altering mutations localized in the Abruptex domain [102]. *Abruptex* was considered as gain of function because its wing phenotype was the opposite to that of loss of function [103]. However, the ectopic activity of Abruptex was suggested [104], and subsequent studies showed that Abruptex phenotypes can be explained by a mild increase in Notch signaling and may have inhibitory effects in some contexts [47].

To date, increasing numbers of cysteine-altering mutations have been reported in fly *Notch* lines [105,106]. Their wing phenotypes were not uniform, including notching, Abruptex, and wild-type (Table 3). It is of note that some CADASIL mutations were reported at the residues corresponding to fly *N* alleles with the Notching or wild-type phenotype but never in those with the Abruptex phenotype (Table 3), which is contradictory to the hypothesis of Fryxell et al.

**Table 3 biomolecules-14-00127-t003:** *Drosophila Notch (N)* EGFr cysteine-altering alleles and wing phenotype.

*N* Allele	Fly Mutation (EGFr)	Wing Phenotype	Residue Corresponding to Human NOTCH3 (EGFr)	CADASIL Mutations Reported at the Residue ^1^
FlyBase [105]				
N^nd−3^	p.Cys105Phe (EGFr 2)	Notching	p.Cys87 (EGFr 2)	p.Cys87Arg [44]/Tyr [44]/Phe [107]
N^Mcd5^	p.Cys739Tyr (EGFr 18)	Wild-type	p.Cys681 (EGFr 17)	not reported
N^Ax−59b^	p.Cys972Gly (EGFr 24)	Abruptex	p.Cys875 (EGFr 22)	not reported
N^Ax-M1^	p.Cys999Tyr (EGFr 25)	Abruptex	p.Cys901 (EGFr 23)	not reported
Yamamoto et al., 2012 [106]				
egf8-C2S	p.Cys343Ser (EGFr 8)	Notching	p.Cys285 (EGFr 7)	p.Cys285Arg [108]
egf8-C2Y	p.Cys343Tyr (EGFr 8)	Notching	p.Cys285 (EGFr 7)	p.Cys285Arg [108]
egf8-C6S	p.Cys369Ser (EGFr 8)	Notching	p.Cys311 (EGFr 7)	p.Cys311Ser [109,110]/Gly [111]
egf9-C5Y	p.Cys398Tyr (EGFr 9)	Notching	p.Cys340 (EGFr 8)	p.Cys340Phe [8]/Trp [112]
egf9-C6S	p.Cys407Ser (EGFr 9)	Notching	p.Cys349 (EGFr 8)	not reported
egf10-C2S	p.Cys413Ser (EGFr 10)	Notching	p.Cys355 (EGFr 9)	p.Cys355Ser ^2^ [113]
egf11-C1S	p.Cys453Ser (EGFr 11)	Notching	p.Cys395 (EGFr 10)	p.Cys395Arg [44]
egf13-C2S	p.Cys535Ser (EGFr 13)	Wild-type	p.Cys478 (EGFr 12)	p.Cys478Tyr [114]
egf25-C2S	p.Cys993Ser (EGFr 25)	Notching	p.Cys896 (EGFr 23)	not reported
egf29-C2S	p.Cys1155Ser (EGFr 29)	Abruptex	p.Cys1055 (EGFr 27)	not reported
egf34-C1Y	p.Cys1341Tyr (EGFr 34)	Wild-type	p.Cys1250 (EGFr 32)	p.Cys1250Trp [115]/Gly ^2^ [113]

^1^ CADASIL mutations were identified using HGMD Professional (2023.3) [54]. ^2^ p.Cys355Ser and p.Cys1250Gly are described by genome position (GRCh38) 19:15189401_C/G and 19:15178912_A/C, respectively.

Using *Drosophila*, lateral inhibition, inductive signaling, asymmetric cell division, and intracellular Notch trafficking can be analyzed in vivo. Nurmahdi et al. focused on 19 Notch missense mutations, 9 of which were cysteine-altering mutations in EGFr [116]. As summarized in Table 4, defects of signaling and trafficking tend to be milder as the mutant EGFr locates to the C-terminal, in other words, increasing the numbering of EGFr. This is consistent with the EGFr location-phenotype correlation reported in CADASIL (see Section 2.5.4). Although all mutations showed a defect in bristle formation, some mutations showed an impaired effect, and others were normal regarding lateral inhibition and inductive signaling (Table 4). These diverse effects and wing phenotypes (Table 3) may be due to the location of mutation (EGFr) and/or context-dependent role of Notch signaling. However, whether each mutation reflects the pathophysiology of CADASIL remains unknown. To our knowledge, the accumulation of the NECD in extracellular spaces has not been reported in *Drosophila*. Thus, the possibility of GOM-like deposits and their association with Notch signaling in the fly remain unknown.

**Table 4 biomolecules-14-00127-t004:** Notch signaling and intracellular trafficking in *Drosophila Notch (N)* EGFr cysteine-altering alleles.

N Allele	Mutation (EGFr)	Bristle Formation ^1^	Lateral Inhibition ^2^	Inductive Signaling ^3^	Intracellular Trafficking (Localization)
N^X^	p.Cys343Ser (EGFr 8)	Absent	Neurogenic	Depletion	Abnormal (loss of expression)
N^Omicron^	p.Cys343Tyr (EGFr 8)	Absent	Neurogenic	Depletion	Abnormal (ER ^4^)
N^Gamma^	p.Cys398Tyr (EGFr 9)	Absent	Neurogenic	Depletion	Abnormal (ER)
N^S^	p.Cys407Ser (EGFr 9)	Absent	Neurogenic	Depletion	Abnormal (ER)
N^Iota^	p.Cys413Ser (EGFr 10)	Absent	Neurogenic	Depletion	Abnormal (ER)
N^G^	p.Cys535Ser (EGFr 13)	Absent	Brain deformation	Depletion	Normal
N^Zeta^	p.Cys993Ser (EGFr 25)	Absent	Neurogenic	Depletion	Abnormal (ER)
N^H^	p.Cys1155Ser (EGFr 29)	Absent	Normal	Normal	Abnormal (Early endosomes)
N^J^	p.Cys1341Tyr (EGFr 34)	Absent	Normal	Normal	Normal

Modified from Nurmahdi et al., 2022 [116]. ^1^ Defects of bristle formation reflect defects in lateral. inhibition and asymmetric cell division. ^2^ Defects of lateral inhibition in the embryonic central nervous system are neurogenic. Brain deformation reflects region-specific impairment of lateral inhibition. ^3^ Inductive signaling was evaluated by boundary cell formation in the embryonic hindgut. ^4^ ER: endoplasmic reticulum.

Taken together, it may be difficult to determine which allele of fly *Notch* corresponds to CADASIL. However, studies of *Drosophila* Notch may be helpful to clarify the function of CADASIL-causing mutations, because EGFr-specific function and interactions between EGFrs in vivo have been intensively studied in *Drosophila* [47].

## 5. Notch Signaling in CADASIL

### 5.1. Biological Role of NOTCH3 Signaling in Vessels

Following the identification of *NOTCH3* as a causative gene for CADASIL, not only the pathological but also the biological role of NOTCH3 in vessels have been focused on. The primary targets of CADASIL are VSMCs of small arteries and pericytes of capillaries [13,24]. Immunohistochemical and in situ hybridization analysis showed that, in the post-developmental stage, NOTCH3 is normally expressed in VSMCs in arteries and also pericytes in capillaries [24,26,117]. *Notch3* knockout mice were fertile, viable, and normally developed [118] but did not show a CADASIL-like phenotype including GOM [25,119,120]. However, detailed analysis revealed abnormal patterning of the cerebrovascular structure, impaired postnatal differentiation, and myogenic tone of VSMCs, suggesting roles of Notch3 in postnatal vasculature differentiation and VSMC maturation [25,121]. Thinning and disorganization of the tunica media of the tail and cerebral arteries [122] and increased susceptibility to brain ischemia [123] were also reported. Transcriptome analysis identified Notch3 target genes, robustly downregulated in distal arteries of Notch3-null mice [124]. In addition, temporal changes in α-smooth muscle cell actin (SMA) staining of the retinal artery in *Notch3* knockout mice until 20 postnatal days showed progressive loss of VSMCs, suggesting an important role of Notch3 in VSMC survival [125]. Some patients with bi-allelic loss-of-function mutations of *NOTCH3* were reported [55,56,57]. The patients had some CADASIL-like features, including arteriopathy, leukoencephalopathy, and stroke, but they were distinct from CADASIL patients because of child onset, no GOM, and other characteristic features. One patient had a homozygous nonsense *NOTCH3* mutation, p.Cys966Ter, and was initially diagnosed with Sneddon syndrome [55]. The expression levels of some of the NOTCH3 target genes identified in Noch3-null mice [124] were decreased in skeletal muscle tissues of the patient and, notably, also CADASIL patients [55], suggesting decreased NOTCH3 signaling in CADASIL. Two patients from another family were similar to the first case in having a homozygous nonsense mutation, p.Arg735Ter, and Sneddon syndrome [56]. Other patients did not have Sneddon syndrome and had the homozygous frameshift mutation p.Arg10Hisfs*16 or compound heterozygous mutations, p.His1944Tyr and p.Leu1976Profs*11 [57]. p.His1944Tyr is located in ANK in NICD, considered as loss of function because His1944 is conserved in the ALHWAAAVNN motif across vertebrates [57].

Although *Notch3* knockout mice did not show GOM, their VSMC phenotype exhibited some similarities with CADASIL model mice and *NOTCH3*-related patients. In tail arteries of *Notch3* null mice, VSMCs showed an irregular shape and smaller size and did not reach their appropriate place and orientation, leading to a disorganized tunica media composed of non-cohesive VSMCs, accompanied by an enlargement of VSMC intercellular spaces [121]. Retinal arteries in *Notch3* knockout mice showed a progressive loss of VSMCs and thickened basement membranes [125]. Immature tail arteries at birth were indistinguishable between *Notch3* knockout and wild-type mice, suggesting an impairment in the postnatal maturation of VSMCs in mice. These VSMC phenotypes, including a decreasing number of VSMCs, morphological change and disorientation of VSMCs, and enlargement of VSMC intercellular spaces and/or thickened basement membranes, appear to overlap with those in a patient with bi-allelic loss-of-function *NOTCH3* mutations [55], CADASIL patients [12,55], and *Notch3* R90C transgenic mice [71]. Taken together, the impairment of VSMC maturation or maintenance of the maturation state, at least partially, may contribute to the pathophysiology of CADASIL.

### 5.2. NOTCH3 Signaling Process in CADASIL Pathophysiology

#### 5.2.1. NOTCH3 Signaling Activity in CADASIL

CADASIL-causing mutations localize in EGFr of N3ECD, indicating that N3ICD is intact in CADASIL patients. Most mutations localized outside of the ligand-binding domain EGFr 10–11 and showed normal ligand binding and Notch signaling activity in cellular experiments [33,53,68,126]. On the other hand, relatively rare mutations in the ligand-binding domain, including p.Cys428Ser in EGFr 10 [53] and p.Cys455Arg in EGFr 11 [126], showed decreased ligand binding and Notch signaling activities. Stroke susceptibility of *Notch3* knockout mice was rescued by VSMC-specific expression of the p.Arg1031Cys (R1031C) transgene but not rescued by the p.Cys455Arg (C455R) transgene, indicating decreased Notch signaling of p.Cys455Arg in vivo [75] Recently, Hack et al. investigated the NOTCH3 signaling activity of mutant NOTCH3 using NIH 3T3 cells. They reported lower signaling activity in mutations, both inside and outside of EGFr 10–11, compared with the wild-type [46].

In contrast to cellular experiments, Baron-Menguy et al. suggested increased Notch signaling in a mouse model of CADASIL. They noted a mild, 1.4–1.7-fold change but a significant increase in the expression of Notch3 target genes in TgNotch3R169C (line 88) compared with TgNotch3WT (line 92). In addition, the genetic reduction in Notch3 signaling by conditional specific knockdown of *Rbpj* in VSMC ameliorated the reduction in the passive diameter of cerebral arteries of TgNotch3R169C [78].

It is noteworthy that Kofler et al. reported the presence of GOM deposits in *Notch3*−/−; *Notch1*+/− double transgenic mice but not in *Notch3*−/− mice [127]. The secreted mutant N3ECD may bind to other vascular Notch family members, resulting in the blocking of their signaling. As both Notch1 and 3 are expressed in VSMCs, Notch1 may compensate for the dysfunction of mutant Notch3 in vivo. In *Notch3*−/−; *Notch1*+/− mice, a decreased dose of *Notch1* may be insufficient to compensate for total Notch signaling activity. Their finding that GOMs form in *Notch3*−/−; *Notch1*+/− can be interpreted as follows: very low total Notch signaling activity in VSMCs causes GOMs (in this case, Notch3-negative GOMs). Their finding suggests that N3ECD-positive GOMs in CADASIL may be a consequence of total Notch signaling dysfunction in VSMCs [127].

#### 5.2.2. Transendocytosis of N3ECD

Previous pathological studies detected GOM in most mutations, including signal-defecting ones, p.Cys428Ser [21,74], and p.Cys455Arg [75]. This suggests that mutant N3ECD accumulates and leads to GOM deposits, regardless of signaling activity. However, the mechanism of the extracellular accumulation of mutant N3ECD remains unknown. By coculturing a stable *NOTCH3*-HEK293 cell line and *JAG1*-HEK293 cell line, Watanebe-Hosomi et al. noted a delayed degradation of p.Cys185Arg mutant N3ECD compared with the wild-type on the cell surface and hypothesized the impairment of transendocytosis in mutant N3ECD [128]. Based on this hypothesis, mutant N3ECD with attenuated ligand-binding activity is considered to accumulate more slowly than N3ECD with common mutations. This is in agreement with the finding that patients with EGFr 10–11 mutations were associated with a relatively preserved cognitive function compared with those with common mutations [74]. In contrast, by coculturing stable *NOTCH3*-HEK293 and *JAG1*-NIH 3T3 cell lines, Suzuki et al. reported that transendocytosis was intact in p.Cys185Arg, whereas it was fully impaired in p.Cys428Ser [129]. They concluded that the accumulation of p.Cys185Arg may not be related to transendocytosis and that the accumulation of p.Cys428Ser N3ECD may be due to impaired transendocytosis. The reason for these contradictory findings remains unclear, but it may be partly due to differences in experimental conditions.

#### 5.2.3. Cis-Interaction

Cis-interaction must be addressed, because both NOTCH3 and its ligand JAG1 are expressed in VSMCs. Cis-interaction is commonly known to inhibit Notch signaling through cis-inhibition of Notch by its ligand or cis-inhibition of its ligand by Notch [130]. Recent studies also reported the possibility of cis-activation [131,132]. Aberrant Notch signaling activities of NOTCH3 mutations, previously reported (see Section 5.2.1), might be at least partially due to impaired cis-interaction. However, to our knowledge, the involvement of cis-interaction between NOTCH3 and JAG1 in CADASIL pathophysiology has not been reported to date.

#### 5.2.4. Glycosylation of N3ECD

Studies focusing on the post-translational processing of CADASIL-causing mutations are limited. Arboleda-Velasquez focused on glycosylation and reported that the CADASIL-causing mutation impaired Fringe-mediated *O*-fucose elongation on NOTCH3 using CHO Lec 1 cells co-transfected with *NOTCH3* EGFr 1–5 and *Fringe* [133]. Suzuki et al. reported that the lunatic fringe may increase the aggregation propensity of mutant NOTCH3 using HeLa cells co-transfected with full-length *NOTCH3* and *lunatic fringe* [129].

## 6. Protein Accumulation or Aggregation in CADASIL

### 6.1. TIMP3 and VTN

Monet-Lepretre et al. performed proteomic analysis of an N3ECD-enriched fraction prepared from frozen brain tissues of one CADASIL patient and one control individual, and they identified 104 proteins enriched in CADASIL [134]. Of the 104 proteins, they focused on Metalloproteinase inhibitor 3 (TIMP3) and Vitronectin (VTN), both of which are functionally important extracellular matrix (ECM) proteins. TIMP3 and VTN were also present in brain artery samples of NOTCH3 R90C transgenic mice but absent in NOTCH3 wild-type transgenic mice [134]. Immunohistochemical analysis of the CADASIL brain confirmed the co-localization of TIMP3 and VTN with N3ECD-positive aggregates. Through electron microscopic analysis, they successfully detected TIMP3 immunogold labeling in GOM, whereas a suitable antibody was not available for VTN. Through further cellular experiments and an immunoprecipitation assay, they showed that the aggregation of NE3CD enhanced N3ECD-TIMP3 complex formation, and TIMP3 promoted complex formation including N3ECD and VTN. These observations support the N3ECD cascade hypothesis that N3ECD accumulation initiates GOM formation by recruiting ECM proteins such as TIMP3, which recruits additional proteins such as VTN [3]. In other words, mutant N3ECD is considered to accumulate first, followed by TIMP3 and VTN, in this order. The results of analysis of the genetic reduction in TIMP3 or VTN in TgNotch3^R169C^ by Capone et al. were compatible with this hypothesis [135]. In TgNotch3^R169C^ mice, cerebral blood flow (CBF) deficit was detected at 5–6 months, much earlier than pathological white matter lesions at 18–20 months (Table 2) [72,78]. The CBF deficits, including impaired CBF responses to neural activity (functional hyperemia) and impaired CBF autoregulation, were ameliorated by the genetic reduction in *Timp3* but not in *Vtn*. In contrast, as a late deficit, the number of pathological white matter lesions was decreased by the genetic reduction in *Vtn* but not in *Timp3*. In TgNotch3^R169C^ mice, N3ECD accumulation and GOM deposition were observed at approximately the same stage as the detection of a CBF deficit. It was notable that both the numbers of N3ECD deposits and GOM in TgNotch3^R169C^ were unaffected by the genetic reduction in *Timp3* or *Vtn*, supporting the suggestion that N3ECD accumulation occurs upstream of GOM [135]. Transgenic mice with an increased expression of human *TIMP3* mimicked the impaired vascular function phenotype of CADASIL transgenic mice [135]. Taken together, these findings suggest that accumulations of TIMP3 and VTN have divergent influences on CADASIL: TIMP3 on CBF deficit in the early stage and VTN on pathological WML in the late stage [135].

### 6.2. LTBP-1 and HTRA1

Cerebral autosomal recessive arteriopathy with subcortical infarcts and leukoencephalopathy (CARASIL) is a hereditary cerebral small vessel disease caused by mutation of *high-temperature requirement A serine peptidase 1 (HTRA1)*, being extremely rare and identified mainly in East Asia [136]. Transforming growth factor-β (TGF-β) is a multifunctional cytokine, and its signaling plays important roles in vascular development and disease [137]. HTRA1 cleaves pro-TGF-β, and the cleaved pro-TGF-β is degraded by the endoplasmic reticulum-associated degradation system [138]. Therefore, loss-of-function mutation *HTRA1* results in the stimulation of TGF-β signaling, which is considered to be a key pathophysiological mechanism of CARASIL and *HTRA1*-related cerebral small vessel disease [139].

Kast et al. focused on ECM proteins related to the TGF-β pathway, and they found that latent TGF-β binding protein 1 (LTBP-1) was strongly colocalized with N3ECD via analyzing brain tissues of CADASIL patients by immunostaining. The co-aggregation properties of N3ECD and LTBP-1 were confirmed by in vitro assays [88]. They suggested a role of dysregulated TGF-β signaling in CADASIL, although whether the signaling is increased or decreased remains unknown.

Zellner et al. performed proteome analysis of brain vessels from autopsied samples of six CADASIL patients and six controls. They identified 95 proteins showing increased abundance in patients. They noted a marked increase in the HTRA1 content, at 4.7-fold, in the patient group. They also confirmed that HTRA1 colocalized with N3ECD deposits in the affected vessels. These findings suggest that HTAR1 is sequestrated in abnormal aggregation, resulting in a loss of its protease activity, which is supported by an increase in HTRA1 substrates in the proteome of patients [89].

Klose et al. investigated the interaction of NOTCH3 and HTRA1 in CADASIL [140,141]. They showed that Jagged-1 is a substrate of HTRA1, and *HTRA1*-silencing in human umbilical artery smooth muscle cells (HUASMCs) resulted in an increase in *Jagged-1* expression, considered as NOTCH3 signaling activity. Their finding is in agreement with a previous report of increased NOTCH3 signaling in CADASIL by Baron-Menguy et al. (see Section 5.2.1) [78]. Clarifying the crosstalk between NOTCH3 and HTRA1, in other words, CADASIL and CARASIL, is an attractive issue that warrants further investigation.

## 7. Conclusions

Researchers have intensively examined the pathophysiology of CADASIL (Figure 5). The effects of mutant N3ECD on NOTCH3 signaling have been studied, but the results differ among reports, partly due to differences in assay conditions. The hypothesis of ECM protein accumulation following mutant N3ECD accumulation is the most commonly adopted; however, the process prior to N3ECD accumulation still remains unknown. In addition, this review cannot support the suggestion that N3ECD accumulation and GOM deposits are the sole cause of VSMC death or that increased break down of the endothelial barrier and leakage in CADASIL lead to the infarcts. To clarify these points, a more appropriate model of CADASIL is necessary.

## Figures and Tables

**Figure 1 biomolecules-14-00127-f001:**
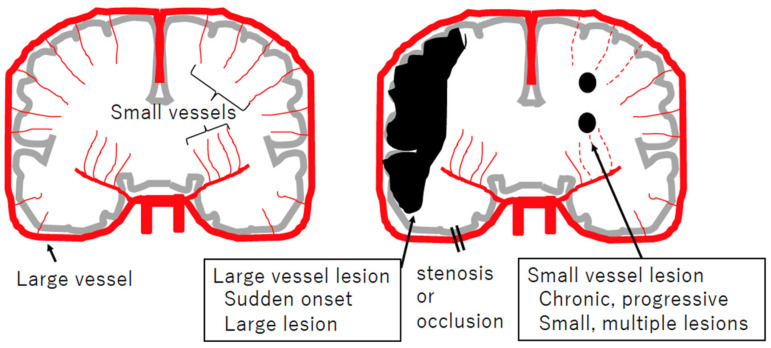
Schematic view of large and small vessel diseases, focusing on cerebral infarction. Coronary sections of the brain (gray), arteries (red), and ischemic lesions (black) are illustrated. Large artery obstruction usually occurs suddenly and results in ischemia in that region. Dysfunction of small arteries is usually chronic and progressive, resulting in small and multiple lesions, referred to as lacunar infarctions.

**Figure 3 biomolecules-14-00127-f003:**
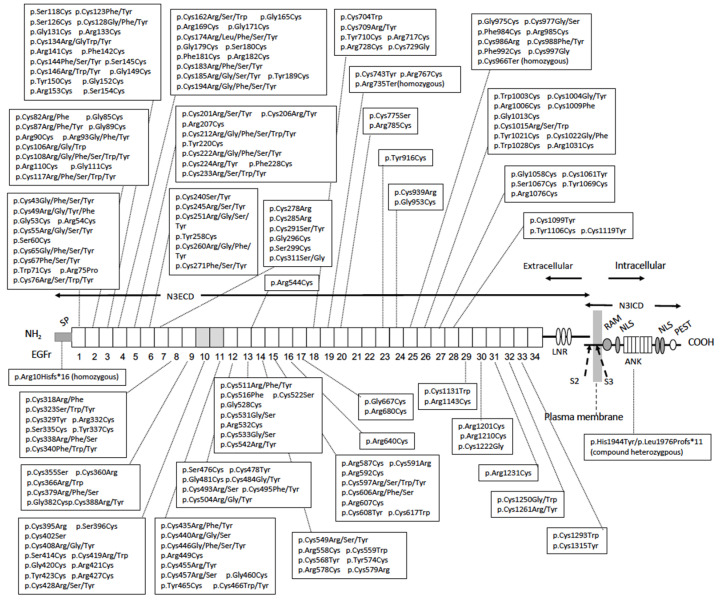
Structure of NOTCH3 receptor protein. NOTCH3 receptor is a non-covalent heterodimer of the extracellular domain (N3ECD) and intracellular domain (N3ICD). N3ECD contains the signal peptide (SP), 34 epidermal growth factor-like repeats (EGFr), of which the 10–11th ones are the ligand binding domain (shaded) and Lin-12/Notch repeat (LNR). N3ICD contains the RBPjκ association module (RAM), ankyrin repeat (ANK), and nuclear localizing sequence (NLS), and is rich in proline (P), glutamate (E), serine (S), and threonine (T) (PEST) domains. The S2 cleavage site (S2) by “a disintegrin and metalloproteinases” (ADAM) protease and intramembrane S3 cleavage site (S3) by γ-secretase are shown. *NOTCH3* mutations reported in CADASIL (Section 2.5 and HGMD Professional 2023.3 [54]) and bi-allelic loss-of-function mutations reported in autosomal recessive early-onset arteriopathy (see Section 5.1) [55,56,57] and their localizations are shown.

**Figure 4 biomolecules-14-00127-f004:**
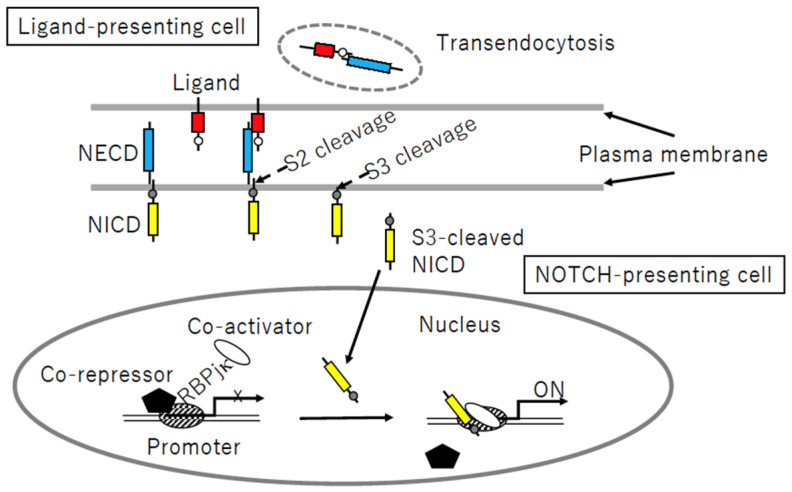
Notch signaling process. Following receptor–ligand binding, sequential S2- and S3-cleavage of the Notch intracellular domain (NICD, yellow) allows the cleaved protein to enter the nucleus, whereas the (red) ligand–Notch extracellular domain (NECD, blue) complex is transendocytosed into a ligand-presenting cell. Nuclear NICD is associated with the DNA-binding protein RBPjk, followed by detachment of the co-repressor and recruitment of the co-activator, resulting in target gene activation.

**Figure 5 biomolecules-14-00127-f005:**
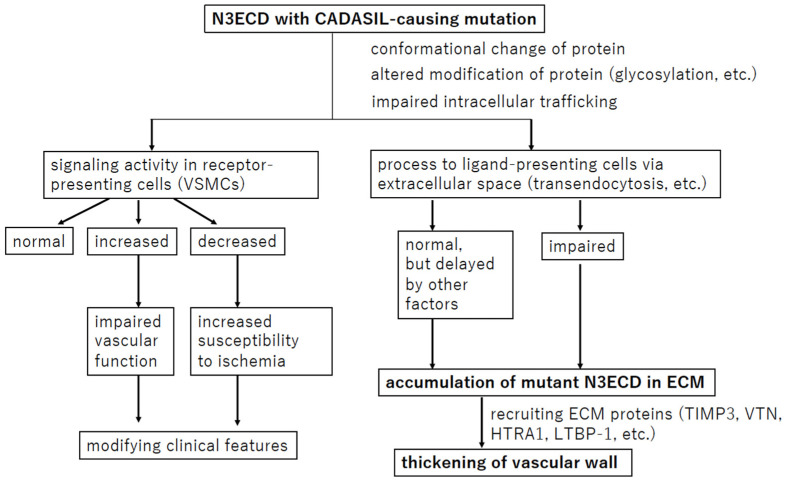
Overview of the hypothesis regarding the pathophysiology of CADASIL. The most widely held hypothesis and confirmed events are in bold.

**Table 1 biomolecules-14-00127-t001:** Materials and applications to clarify the pathophysiology of CADASIL.

Human materials: autopsied brain tissue or vessels, skin biopsy specimens	
	Histopathology, immunohistochemistry
	Biochemistry, proteomics
	Gene expression, transcriptome
Animal models: transgenic mice	
	Genetic approach to clarify pathophysiology
	Temporal analysis of disease process
	Histopathology, immunohistochemistry
	Biochemistry, proteomics
	Gene expression, transcriptome
Cell cultures: VSMCs ^1^, iPS cell-derived mural cells, cell lines (HEK293, NIH 3T3, etc.)	
	Analysis of Notch signaling activity
	Recreation of pathology of CADASIL
	Vaiability and proliferation
In vitro	
	Aggregation assay of N3ECDpeptides

^1^ vascular smooth muscle cells.

## Data Availability

No new data were created or analyzed in this study. Data sharing is not applicable to this article.

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
