# Peer review of "Progress to Clarify How NOTCH3 Mutations Lead to CADASIL, a Hereditary Cerebral Small Vessel Disease"

_biomolecules, 2024, doi:10.3390/biom14010127_

Round 1

Reviewer 1 Report

Comments and Suggestions for Authors

The authors aimed to provide an updated review into the current understanding into the causes of CADISIL as it relates to Notch3. However, their premise that CADASIL is caused by abnormal accumulation of N3ECD in the extracellular space and not the loss of Notch3 signaling is not supported by the literature. Moreover, the English writing and grammar are very poor. The current manuscript is tedious to read and hard to follow. The authors often go off on tangents without providing enough information to understand how the information relates to CADASIL or its pathobiology. 

Major Comments: 

·       The abstract is too focused on the role of N3ECD accumulation in the extracellular space in CADASIL. Studies of Notch3 knock-out mouse models clearly demonstrate a role for cell autonomous Notch3 signaling in vascular smooth muscle cell maturation followed by cell death in the adult mouse. This latter is minimized (only one paragraph in the middle of the paper) and does not include all the papers published on mouse models as it relates to CADASIL.

·       Section 5.1, Biological role of NOTCH3 signaling in vessels, is incomplete and missing a significant amount of literature.  Below is some, but not all, the literature that should be incorporated.

1)     GOM formation has been shown in Notch3 null mice – PMID: 26563570

2)     Mice with Jag1 deletion in VMSCs also developed GOMs. PMID: 21068062

·       There are human cases of CADASIL in which reduced Notch3 expression is seen with specific Notch3 variants without GOMs development, such as PMID: 25870235.  A discussion of these mutations should be incorporated as it correlates with the mouse models. 

·       Notch3 structure and signaling (section 3) should be presented prior to discussing the human CADASIL mutations.  It would also be helpful to make a figure in which the known human variants and their location in the protein or EGFrs is shown. This could be added to Figure 3.

·       There are similarities in the VSMC phenotypes between the CADASIL Knockin mouse models and the Notch3 loss of function models that should be discussed.

·       The statement,  “it remains unclear whether NOTCH3-JAG1 interaction occurs between endothelial cells and VSMCs, or between neighboring VSMCs” is inaccurate.  Mouse studies suggest Jag1 is necessary in both ECs and VSMCs and both signal to Notch3 in the VSMCs. Endothelial specific Jag1 deletion mouse studies demonstrated similar  VSMC phenotype as that of loss of Notch3.  This strongly suggests that Jag1(EC)-Notch3 (VMSC) signaling. Loss of VSMC Jag1 also leads to VSMC defects as well as EC defects.  This latter model developed GOMs again suggesting a loss of Jag1/N3 signaling contributes to the phenotype. This could occur in two ways, by the secreted N3ECD forming inactive complexes with Jag1 and blocking endogenous Notch signaling and/or a loss of cell autonomous Notch3 signaling. 

·       The section on Drosophila Notch does not add to the review.  Drosophila Notch is more similar to Notch1 and Notch 2, than Notch3 as it has different numbers of EGFrs.  The authors even state at the end of this section. “However, whether each mutation reflects the pathophysiology of CADASIL remains unknown. Taken together, it may be difficult to determine which phenotype or allele of fly Notch corresponds to CADASIL.” This section should be removed. 

·       The current manuscript does support a link between GOM and N3ECD and the protein aggregates that arise. However, the literature cited does not support that N3ECD and GOMs are the sole cause the VSMC death and increased break down of the endothelial barrier and leakage in CADASIL that leads to the infarcts.

Minor Comments:

·       Figure 2 uses images from prior publications. This may require copyright permission. 

·       The use of the term “Notch receptors” is misleading as it suggests that Notch is the ligand for the receptor, but it not.  Notch proteins, or Notch family members is a more appropriate term to use. 

·       The statement that “Each EGFr normally contains six cysteine residues that are considered to form three pairs of disulfide bonds to stabilize the protein conformation” needs to be better described.  It should be pointed out that cysteines in one repeat form disulfide bonds with cysteines from  neighboring repeat domains. Thus, CADASIL variants disrupt not only the repeat the variant is within, but the neighboring repeat domain as well. 

·       The authors should discuss how protein misfolding leads to release of the Notch3 extracellular domain in the section, 2.5.2. 

·       Abbreviations are not defined at their first use.  NICD  is used in section 3.1 prior to its defining in section 3.2.  

Comments on the Quality of English Language

The overall English grammar, sentence structure and paragraph structure are poor. Ideas are off mentioned without context presented. The review also needs to be more focused and tangential sections that don't add to the topic of the review removed or minimized. 

Author Response

We thank the reviewer for taking the time to review this manuscript. We have read his/her comments carefully.

First paragraph of the comments
The authors aimed to provide an updated review into the
current understanding into the causes of CADISIL as it
relates to Notch3. However, their premise that CADASIL is
caused by abnormal accumulation of N3ECD in the
extracellular space and not the loss of Notch3 signaling is
not supported by the literature. Moreover, the English
writing and grammar are very poor. The current manuscript
is tedious to read and hard to follow. The authors often go
off on tangents without providing enough information to
understand how the information relates to CADASIL or its
pathobiology.

According to the reviewer’s criticism, we revised the manuscript as follows.
We respond to the criticism of English writing at the end of this letter.

Major comment 1
· The abstract is too focused on the role of N3ECD
accumulation in the extracellular space in CADASIL.
Studies of Notch3 knock-out mouse models clearly
demonstrate a role for cell autonomous Notch3 signaling in
vascular smooth muscle cell maturation followed by cell
death in the adult mouse. This latter is minimized (only one
paragraph in the middle of the paper) and does not include
all the papers published on mouse models as it relates to
CADASIL.

Response 1
As the reviewer commented, Notch3 knockout models revealed that Notch3 is essential for the development and differentiation of VSMCs. Previously, most studies reported that NOTCH signaling was intact in the presence of CADASIL mutations; however, reports of decreased signaling in their presence have been increasing to date. According to the reviewer’s criticism, we softened the phrase in Abstract. We added description of knock-out mice in 5.1. (please see response 2).

Major comment 2
· Section 5.1, Biological role of NOTCH3 signaling in
vessels, is incomplete and missing a significant amount of
literature. Below is some, but not all, the literature that
should be incorporated.
1) GOM formation has been shown in Notch3
null mice – PMID: 26563570
2) Mice with Jag1 deletion in VMSCs also
developed GOMs. PMID: 21068062

Response 2
We added a description of Notch3 null mice to section 5.1. (lines 446-449), and cited the following reference:
[118] Belin de Chantemele et al., 2008   PMID: 18818417
[119] Arboleda-Velasquez et al., 2008  PMID: 18347334
[120] Fouillade et al., 2013   PMID: 23117660

lines 446-449
Thinning and disorganization of the tunica media of the tail and cerebral arteries [118] and increased susceptibility to brain ischemia [119] were also reported. Transcriptome analysis identified Notch3 target genes, robustly downregulated in distal arteries of Notch3-null mice [120].

We feel that it is difficult to add the two references recommended by the reviewer.
In PMID: 26563570, the authors stated that GOM was detected in Notch1+/-, Notch3-/- double transgenic mice, not in Notch3-/- mice. We think that GOM in Notch1+/-, Notch3-/- mice should more precisely be referred to as GOM-like deposits, because N3ECD is not included in the deposits.  
In PMID: 21068062, we failed to find a description of GOM; however, we cite this in response to the reviewer’s comment 6.

Major comment 3
· There are human cases of CADASIL in which reduced
Notch3 expression is seen with specific Notch3 variants
without GOMs development, such as PMID: 25870235. A
discussion of these mutations should be incorporated as it
correlates with the mouse models.

Response 3
Patients with bi-allelic NOTCH3 loss-of-function mutations were described in three reports, including PMID: 25870235. These patients are clinically distinct from CADASIL patients, although they exhibit CADASIL-like features including arteriopathy, leukoencephalopathy, and stroke. Some Notch3 target genes identified in Notch3 null mice were also downregulated in these patients. We added this discussion to section 5.1., following the description of Notch3 null mice.
We also added the following reference:
[55] Pippucci et al., 2015   PMID: 25870235[56] Greisenegger et al., 2021  PMID: 32980981
[57] Stellingwerff et al., 2022  PMID: 35026854
[120] Fouillade et al., 2013   PMID: 23117660

lines 452-465
Some patients with bi-allelic loss-of-function mutations of NOTCH3 were reported [55-57]. The patients have some CADASIL-like features, including arteriopathy, leukoenceph-alopathy, and stroke, but they were distinct from CADASIL patients because of child-onset, no GOM, and other characteristic features. One patient had homozygous nonsense NOTCH3 mutation, p.Cys966Ter, and was initially diagnosed with Sneddon syndrome [55]. Expression levels of some of the NOTCH3-target genes identified in Noch3-null mice [120] were decreased in skeletal muscle tissues of the patient, and notably, also CADASIL patients [55], suggesting decreased NOTCH3 signaling in CADASIL. Two patients from another family were similar to the first case in having homozygous non-sense mutation, p.Arg735Ter, and Sneddon syndrome [56]. Other patients did not have Sneddon syndrome and had the homozygous frameshift mutation p.Arg10Hisfs*16, or compound heterozygous mutations, p.His1944Tyr and p.Leu1976Profs*11 [57]. p.His1944Tyr is located in ANK in NICD, considered as loss-of-function because His1944 is conserved in the ALHWAAAVNN motif across vertebrates [57].

Major comment 4
· Notch3 structure and signaling (section 3) should be
presented prior to discussing the human CADASIL
mutations. It would also be helpful to make a figure in
which the known human variants and their location in the
protein or EGFrs is shown. This could be added to Figure
3.

Response 4
We understand that the reading of section 3, especially section 3.1., would be helpful prior to reading section 2.5.; however, we prefer to describe the clinical and genetic aspects of CADASIL together, and not interrupt them by inserting  section 3. Alternatively, we have inserted text (see 3.1. and Figure 3) in section 2.5.2., the first section discussing CADASIL mutations (line 158). 
In response to the latter comment and to prevent Figure 3 from becoming too busy, a summary of known CADASIL mutations and bi-allelic loss-of-function mutations reported in early-onset arteriopathy with their locations was added to Figure 3 (Figure 3 and legend: lines 240-242).

Major comment 5
· There are similarities in the VSMC phenotypes between
the CADASIL Knockin mouse models and the Notch3 loss
of function models that should be discussed.

Response 5
Unfortunately, we failed to find similarities in VSMC phenotypes between the CADASIL knock-in mouse models and Notch3 knockout models in the literature; therefore, we did not add them to the discussion. Although the degeneration of VSMCs was observed in both models, GOM was noted only in knock-in mice. Physiological analysis of isolated cerebral arteries showed a significant decrease of the passive diameter with increases of blood pressure in the physiological range in R170C knock-in mice (Baron-Menguy et al., 2017; PMID: 27821617), whereas the passive diameter remained unchanged in Notch3 null mice (Belin de Chantemele et al., 2008; PMID: 18818417). 

Major comment 6
· The statement, “it remains unclear whether NOTCH3-
JAG1 interaction occurs between endothelial cells and
VSMCs, or between neighboring VSMCs” is
inaccurate. Mouse studies suggest Jag1 is necessary in
both ECs and VSMCs and both signal to Notch3 in the
VSMCs. Endothelial specific Jag1 deletion mouse studies
demonstrated similar VSMC phenotype as that of loss of
Notch3. This strongly suggests that Jag1(EC)-Notch3
(VMSC) signaling. Loss of VSMC Jag1 also leads to VSMC
defects as well as EC defects. This latter model developed
GOMs again suggesting a loss of Jag1/N3 signaling
contributes to the phenotype. This could occur in two ways,
by the secreted N3ECD forming inactive complexes with
Jag1 and blocking endogenous Notch signaling and/or a
loss of cell autonomous Notch3 signaling.

Response 6
We appreciate the reviewer pointing out our inaccurate statement. We corrected it following the reviewer’s informative comment (section 3.3., lines 275-285). We cited additional literature outlined below; however, we failed to identify a description of GOM in either of the citations:
[65] High et al., 2008 (PMID: 18245384) and [66] Breikaa et al., 2022 (PMID: 35792302) for EC-specific knockout.
[67] Feng et al., 2010 (21068062) and [68] Basu et al., 2018 (PMID: 30168730) for VSMC-specific knockout of JAG1. 

lines 275-285
Endothelial or VSMC-specific deletion in mice strongly suggest that NOTCH3-JAG1 interaction occurs between endothelial cells and VSMCs, and also between neighboring VSMCs. Mice with endothelial-specific deletion of Jag1 are embryonic-lethal, showing markedly decreased expression levels of VSMC markers but intact expression levels of endothelial markers [65]. Tamoxifen-induced conditional knockout of endothelial Jag1 in adult mice caused loss of Jag1 expression and also decreased expressions of Notch3 and its downstream molecules in neighboring VSMCs [66]. Similarly, mice with VSMC-specific deletion of Jag1 are perinatal-lethal, showing a deficit of VSMC development [67], and conditional knockout of VSMC-specific JAG1 in adults led to reduced expression of myosin light chain kinase and impairment of the arterial contractile function [66].  

Major comment 7
· The section on Drosophila Notch does not add to the
review. Drosophila Notch is more similar to Notch1 and
Notch 2, than Notch3 as it has different numbers of
EGFrs. The authors even state at the end of this section.
“However, whether each mutation reflects the
pathophysiology of CADASIL remains unknown. 
Taken together, it may be difficult to determine which phenotype
or allele of fly Notch corresponds to CADASIL.” This
section should be removed.

Response 7
We included the section on Drosophila Notch because the Notch gene and its signaling were first identified in Drosophila, and cysteine-altering mutations in the fly Notch have been discussed in relation to CADASIL (Fryxell et al., 2001; PMID: 11136906; Yamamoto, 2020; PMID: 31943162; Manini and Pantoni, 2021; PMID: 33464533). Although the reviewer does not consider the Drosophila section important, we respectfully consider it to be essential.    

Major comment 8
· The current manuscript does support a link between
GOM and N3ECD and the protein aggregates that arise.
However, the literature cited does not support that N3ECD
and GOMs are the sole cause the VSMC death and
increased break down of the endothelial barrier and
leakage in CADASIL that leads to the infarcts.

Response 8
As the reviewer commented, N3ECD accumulation and GOM are responsible as a partial but not the sole cause of VSMC death and other pathophysiologies of CADASIL. We added this limitation to Conclusion.

lines 592-596
In addition, this review cannot support the suggestion that N3ECD accumulation and GOM deposits are the sole cause of VSMC death or that increased break down of the endothelial barrier and leakage in CADASIL leads to the infarcts. To clarify these, a more appropriate model of CADASIL is necessary.

Minor comment 1
· Figure 2 uses images from prior publications. This may
require copyright permission.

Response 1
We contacted the publishers and obtained copyright permission before the initial submission, and are sending permission forms to the editorial office. We added a description of copyright permission to the Figure 2 legend. 

lines 125-126
Upper: Axial section of fluid-attenuated inversion recovery (FLAIR) MRI of a patient with CADASIL, reproduced with permission from Mizuno, 2012 [22]; published by the Japanese Society of Neurology, with slight modification.

lines 130-131
Lower: Electron micrograph of skin biopsy specimen, reproduced with permission from Mizuno et al., 2008 [23]; published by the Japanese Society of Internal Medicine, with slight modification.

Minor comment 2
· The use of the term “Notch receptors” is misleading as it
suggests that Notch is the ligand for the receptor, but it
not. Notch proteins, or Notch family members is a more
appropriate term to use.

Response 2
The term “Notch receptors” has often been used in published papers, including the references; however, according to the reviewer’s comment, “Notch receptors” has been changed to Notch family members (section 3.1., line 211 and section 3.3., lines 264, 267).

Minor comment 3
· The statement that “Each EGFr normally contains six
cysteine residues that are considered to form three pairs of
disulfide bonds to stabilize the protein conformation” needs
to be better described. It should be pointed out that
cysteines in one repeat form disulfide bonds with cysteines
from neighboring repeat domains. Thus, CADASIL variants
disrupt not only the repeat the variant is within, but the
neighboring repeat domain as well.

Response 3
We appreciate the reviewer’s helpful suggestion, and improved the description in section 2.5.2.

lines 158-164
Each EGFr normally contains six cysteine residues that are considered to form three pairs of disulfide bonds to stabilize the protein conformation, and possibly form disulfide bonds with cysteines from the neighboring repeat domain. Most pathogenic mutations are cysteine-altering missense ones, resulting in a change in the number of cysteine residues in a certain EGFr to an odd number, disrupting the structure of not only the repeat the variant is within, but also the neighboring repeat domain.

Minor comment 4
· The authors should discuss how protein misfolding
leads to release of the Notch3 extracellular domain in the
section, 2.5.2.

Response 4
According to the reviewer’s comment, a discussion was added to 2.5.2.

lines 166-168
Extracellular accumulation of misfolded N3ECD may be due to exocytosis, aggregation on the plasma membrane, and/or impairment of transendocytosis (see 5.2.2.), although the precise mechanism remains unknown.

Minor comment 5
· Abbreviations are not defined at their first use. NICD is
used in section 3.1 prior to its defining in section 3.2.

Response 5
Based on the reviewer’s comment, NICD was defined as “Notch intracellular domain” at the time of its first use (3.1. line 214 and 3.2. line 245).

Comments on the Quality of English Language
The overall English grammar, sentence structure and
paragraph structure are poor. Ideas are off mentioned
without context presented. The review also needs to be
more focused and tangential sections that don't add to the
topic of the review removed or minimized.

Response
We were sorry to read the comments regarding the English of the original manuscript. The English grammar, sentence structure, and paragraph structure of the original manuscript were edited by a professional medical proofreader who is a native English speaker. We have attached a certificate to show this. We will implement any advice/suggestions regarding further editing that may be necessary. Thank you for your consideration. 

Again, we greatly appreciate the reviewer’s helpful comments.

Reviewer 2 Report

Comments and Suggestions for Authors

This review by Mizuta et al comprehensively covers the biology and pathology of cerebral autosomal dominant arteriopathy with subcortical infarcts and leukoencephalopathy (CADASIL), focusing on Notch3. It is well written and will be informative to the field in its current form. However, there are a couple of questions that need to be clarified, which will probably help improve this review for the readers.

The authors extensively describe about granular osmiophilic material (GOM) deposits and Notch 3 extracellular domain accumulation. Where are these deposits and accumulation observed precisely? Are they located in the extracellular space, on the plasma membrane, and/or in the cytoplasm? EM images in Figure 2 are helpful, but are not interpretable for non-experts. Demarcating and labeling organelles and cellular structures will be appreciated.

The location of GOM deposits and Notch 3 extracellular domain accumulation will probably be critical also to consider its molecular mechanism underlying CADASIL pathogenesis. While the authors discuss trans-endocytosis, potential involvement of cis-interaction between Notch 3 and its ligand(s) is not at all discussed. Could this be a potential mechanism, or is this totally unlikely?

Line 523, "brain tissues of patients"– is this CADASIL or CARASIL patients?

Comments on the Quality of English Language

Quality of English language is largely good, presumably thanks to the language editing service the authors used. But, minor editing by the editorial office would probably further increase the quality.

Author Response

We thank the reviewer for taking the time to review this manuscript. We have read his/her comments carefully.

First paragraph of the comments
This review by Mizuta et al comprehensively covers the
biology and pathology of cerebral autosomal dominant
arteriopathy with subcortical infarcts and
leukoencephalopathy (CADASIL), focusing on Notch3. It is
well written and will be informative to the field in its current
form. However, there are a couple of questions that need to
be clarified, which will probably help improve this review for
the readers.

We appreciate the favorable comments. Based on the helpful points raised, we revised the manuscript as follows: 

Comment 1
The authors extensively describe about granular
osmiophilic material (GOM) deposits and Notch 3
extracellular domain accumulation. Where are these
deposits and accumulation observed precisely? Are they
located in the extracellular space, on the plasma
membrane, and/or in the cytoplasm? EM images in Figure
2 are helpful, but are not interpretable for non-experts.
Demarcating and labeling organelles and cellular structures
will be appreciated.

Response 1
GOM deposits localize in extracellular spaces, specifically within or adjacent to the basement membrane of VSMCs. We added a description of the precise localization of GOM deposits in section 2.2 (line 85) and 2.4. (lines 139-143) and cited additional references ([26] Joutel et al., 2010, [27] Lorenzi et al., 2017, and [28] Yamamoto et al., 2013).

line 85
GOM can be detected by electron microscopic observation of areas within or adjacent to the basement membrane……

lines 139-143
……GOM deposition within or adjacent to the basement membrane of VSMCs or pericytes detected on electron microscopic observation (Figure 2). GOM deposits often localize in infoldings of VSMCs and in close contact with the plasma membrane, but separated from the plasma membrane by a thin electron-lucent halo; however, they are sometimes observed distant from the plasma membrane [26-28].

We also added labeling of VSMCs, the extracellular matrix, and intra-/extra-cellular structures (mitochondria, plasma membrane, and collagen fibers) in Figure 2 and a description to the legend (line 133). 

Comment 2
The location of GOM deposits and Notch 3 extracellular
domain accumulation will probably be critical also to
consider its molecular mechanism underlying CADASIL
pathogenesis. While the authors discuss trans-endocytosis,
potential involvement of cis-interaction between Notch 3
and its ligand(s) is not at all discussed. Could this be a
potential mechanism, or is this totally unlikely?

Response 2
We added a discussion of cis-interaction as section 5.2.3. (lines 505-513). We also cited references ([124] del Alamo et al., 2011, [125] Nandagopal et al, 2019, and [126] Ng et al., 2021).

lines 505-513
5.2.3. Cis-interaction 
Cis-interaction must be addressed, because both NOTCH3 and its ligand JAG1 are expressed in VSMCs. Cis-interaction is commonly known to inhibit Notch signaling through cis-inhibition of Notch by its ligand or cis-inhibition of its ligand by Notch [124]. Recent studies also reported the possibility of cis-activation [125,126]. Aberrant Notch signaling activities of NOTCH3 mutations previously reported (see 5.2.1.) might be at least partially due to impaired cis-interaction. However, to our knowledge, involvement of cis-interaction between NOTCH3 and JAG1 in CADASIL pathophysiology has not been reported to date.

Comment 3
Line 523, "brain tissues of patients"– is this CADASIL or
CARASIL patients?

Response 3
The answer is CADASIL. We revised this part in section 6.2. to “brain tissues of CADASIL patients” (line 568).  

Comments on the Quality of English Language
Quality of English language is largely good, presumably
thanks to the language editing service the authors used.
But, minor editing by the editorial office would probably
further increase the quality.

Response
We thank the reviewer for the favorable comments on the English language quality. We will follow any of the editorial office’s suggestions regarding minor editing.

Again, we appreciate the helpful comments made by the reviewer. 

Reviewer 3 Report

Comments and Suggestions for Authors

The article is a review of works devoted to the study of NOTCH3 mutations leading to cerebral autosomal dominant arteriopathy with subcortical infarcts and leukoencephalopathy. The text is easy to read and the structure of the text is logical. The authors reviewed articles from 1991 to 2023, providing readers with information published over the past 30 years. The review is quite detailed; the authors compiled information from various sources into tables and made a brief conclusion. The authors are experts in brain research. The article may be published.

Author Response

Comments and suggestions
The article is a review of works devoted to the study of
NOTCH3 mutations leading to cerebral autosomal
dominant arteriopathy with subcortical infarcts and
leukoencephalopathy. The text is easy to read and the
structure of the text is logical. The authors reviewed articles
from 1991 to 2023, providing readers with information
published over the past 30 years. The review is quite
detailed; the authors compiled information from various
sources into tables and made a brief conclusion. The
authors are experts in brain research. The article may be
published.

Response
We thank the reviewer for taking the time to review this manuscript, and
appreciate the favorable comments regarding publication.

Round 2

Reviewer 1 Report

Comments and Suggestions for Authors

The authors addressed some of the comments sufficiently in the revised manuscript.  However there are still concerns that need to be addressed. 

Figure 3 is not improved but more confusing. Why is the Notch3 structure presented twice? The table at the bottom is not legible.  The mutations should be incorporated into the image of the Notch3 structure which is fairly standard for presenting such work.  

The VSMC phenotypes in the Notch3 KO mice should be discussed in more detail as the failure of VSMC maturation is seen in both humans and mice. Are there any overlaps between CADASIL Notch3 knockin models.  The discussion should not be limited to the GOMs.

Also, could the secreted N3ECD block signaling of other vascular Notch family members such as Notch1? Maybe this is why the GOMs are seen only when Notch1 is heterozygous in a Notch3 null background.  The fact that GOMs form in the Notch3-/-;Notch1+/- is very relevant and should be incorporated and discussed. 

If Drosphila section is kept, then additional information needs to be presented. What are the outcomes of the mutations? How do they affect lateral inhibition?  How do they affect Notch signaling? Are they associated with GOMs? There is not discussion about how this data relates to CADASIL mutations.  What does a “Notching” phenotype or “Abruptex” phenotype mean? Some mutations are loss of function and some of gain of function. The authors are assuming this is common knowledge.  How do you explain variants that have wildtype phenotypes? This section as is does not add to the review unless it is presented in more depth. 

Comments on the Quality of English Language

None at this time. 

Author Response

Comments from reviewer 1 and responses to them

We thank the reviewer very much for taking the time to review the revised manuscript. 

First paragraph of the comments
The authors addressed some of the comments sufficiently in the revised manuscript.  However there are still concerns that need to be addressed. 

We apologize for the fact that we could not address all the issues. According to the reviewer’s criticism, we revised the manuscript (highlighted in blue), as follows:

Comment 1
Figure 3 is not improved but more confusing. Why is the Notch3 structure presented twice? The table at the bottom is not legible.  The mutations should be incorporated into the image of the Notch3 structure which is fairly standard for presenting such work.  

Response 1
We apologize for the confusing nature of the revised Figure 3. We incorporated the mutations into the image of the NOTCH3 structure. 

Comment 2
The VSMC phenotypes in the Notch3 KO mice should be discussed in more detail as the failure of VSMC maturation is seen in both humans and mice. Are there any overlaps between CADASIL Notch3 knockin models.  The discussion should not be limited to the GOMs.

Response 2
According to the reviewer’s criticism, we read the references carefully and understand that VSMC phenotypes in Notch3 knockout mice, including a decreasing number, morphological change, and disorientation of VSMCs, and enlargement of VSMC intercellular spaces and/or thickened basement membranes, appear to overlap with those in a patient with bi-allelic loss-of-function NOTCH3 mutations, CADASIL patients, and Notch3 R90C transgenic mice. We added this information to section 5.1. (lines 483-497)

Comment 3
Also, could the secreted N3ECD block signaling of other vascular Notch family members such as Notch1? Maybe this is why the GOMs are seen only when Notch1 is heterozygous in a Notch3 null background.  The fact that GOMs form in the Notch3-/-;Notch1+/- is very relevant and should be incorporated and discussed. 

Response 3
We appreciate the reviewer’s helpful suggestion. The secreted mutant N3ECD may bind to other vascular Notch family members, resulting in the blocking of their signaling. As both Notch1 and 3 are expressed in VSMCs, Notch1 may compensate for the dysfunction of mutant Notch3 in vivo. In Notch3-/-;Notch1+/- mice, a decreased dose of Notch1 may be insufficient to compensate for total Notch signaling activity. The fact that GOMs form in Notch3-/-;Notch1+/- , but not in Notch3-/- (Kofler et al., 2015 PMID: 26563570 ) can be interpreted as such: very low Notch signaling activity in VSMCs causes GOM (Notch3-negative GOM). This finding suggests that N3ECD-positive GOMs in CADASIL may be a consequence of total Notch signaling dysfunction in VSMCs.
We added this information to section 5.2.1. (lines 518-527)  

Comment 4
If Drosphila section is kept, then additional information needs to be presented. What are the outcomes of the mutations? How do they affect lateral inhibition?  How do they affect Notch signaling? Are they associated with GOMs? There is not discussion about how this data relates to CADASIL mutations.  What does a “Notching” phenotype or “Abruptex” phenotype mean? Some mutations are loss of function and some of gain of function. The authors are assuming this is common knowledge.  How do you explain variants that have wildtype phenotypes? This section as is does not add to the review unless it is presented in more depth. 

Response 4
We apologize for the insufficient information in the Drosophila section.
Outcomes of Drosophila Notch mutations include impairments of lateral inhibition, inductive signaling, and asymmetric cell division. As the role of canonical Notch signaling is context-dependent, the mutations may affect Notch signaling in a context-dependent manner, suggesting that some mutations are loss- or gain-of-function, whereas others are normal. To our knowledge, accumulation of the Notch extracellular domain in extracellular spaces has not been reported in Drosophila; thus, the possibility of GOM-like deposits and their association with Notch signaling in the fly remain unknown. We consider that studies of Drosophila Notch may be helpful to clarify the function of CADASIL-causing mutations, because EGFr-specific functions and interactions between EGFrs in vivo have been intensively studied in Drosophila. We revised Drosophila section 4.4.2. and Table 4 by adding the above and details of Notching and Abruptex.

lines 399-413
As described above, cysteine-altering mutations in a certain EGFr of NOTCH3 lead to a single phenotype, CADASIL, in humans. Some Drosophila Notch mutations have the same characteristics, but their phenotypes are not uniform (Tables 3 and 4). Notch signaling activity includes lateral inhibition, inductive signaling, and asymmetric cell division [100]. In Drosophila wings, Notch loss-of-function alleles cause thicker wing veins and scalloping (notching) in the wing margin. In contrast, Notch gain-of-function alleles, Abruptex, cause abrupting (shortened) wing veins and large wings [101]. All Abruptex mutations localize in EGFr 24-29, called the Abruptex domain. Of the Abruptex alleles, homozygous lethal alleles, known as lethal-Abruptex, were cysteine-altering mutations. Fryxell et al. hypothesized that human CADASIL corresponds to fly Abruptex because, at that time, known cysteine-altering mutations localized in the Abruptex domain [102]. Abruptex was considered as gain-of-function because its wing phenotype was the opposite to that of loss-of-function [103]. However, ectopic activity of Abruptex was suggested [104], and subsequent studies showed that Abruptex phenotypes can be explained by a mild increase in Notch signaling and may have inhibitory effects in some contexts [47].

Following references were added.
[100] Bray, 2006 PMID:16921404
[101] de Celis and García-Bellido, 1994 PMID:7918096 
[103] de Celis and García-Bellido, 1994 PMID:8138156
[104] Perez et al., 2005 PMID:15804562

lines 425-438
Using Drosophila, lateral inhibition, inductive signaling, asymmetric cell division, and intracellular Notch trafficking can be analyzed in vivo. Nurmahdi et al. focused on 19 Notch missense mutations, 9 of which were cysteine-altering mutations in EGFr [116]. As summarized in Table 4, defects of signaling and trafficking tend to be milder as the mutant EGFr locates to the C-terminal; in other words, increasing the numbering of EGFr. This is consistent with the mutation EGFr location-phenotype correlation reported in CADASIL (see 2.5.4). Although all mutations showed a defect in bristle formation, some mutations showed an impaired effect, and others were normal regarding lateral inhibition and inductive signaling (Table 4). These diverse effects and wing phenotypes (Table 3) may be due to the location of mutation (EGFr) and/or context-dependent role of Notch signaling. However, whether each mutation reflects the pathophysiology of CADASIL remains unknown. To our knowledge, accumulation of the NECD in extracellular spaces has not been reported in Drosophila. Thus, the possibility of GOM-like deposits and their association with Notch signaling in the fly remains unknown.

lines 447-450
Taken together, it may be difficult to determine which allele of fly Notch corresponds to CADASIL. However, studies of Drosophila Notch may be helpful to clarify the function of CADASIL-causing mutations, because EGFr-specific function and interactions between EGFrs in vivo have been intensively studied in Drosophila [47].

Again, we greatly appreciate the reviewer’s critical and helpful comments. 
